



# Porous aerosol in degassing plumes of Mt. Etna and Mt. Stromboli

Valery Shcherbakov[1,2], Olivier Jourdan[1], Christiane Voigt[3,4], Jean-Francois Gayet[1], Aurélien Chauvigne[1], Alfons Schwarzenboeck[1], Andreas Minikin[3,*], Marcus Klingebiel[5], Ralf Weigel[4], Stephan Borrmann[4], Tina Jurkat[3], Stefan Kaufmann[3], Romy Schlage[3], Christophe Gourbeyre[1], Guy Febvre[1], Tatsiana Lapyonok[6], Wiebke Frey[7,**], Sergej Molleker[7], Bernadett Weinzierl[3,8]

[1]Laboratoire de Météorologie Physique, UMR 6016 CNRS/Université Clermont Auvergne, Clermont-Ferrand, France.
[2]LaMP, Institut Universitaire de Technologie d'Allier, Montluçon, France.
[3]Institut für Physik der Atmosphäre, Deutsches Zentrum für Luft- und Raumfahrt (DLR), Oberpfaffenhofen, Germany.
[4]Institut für Physik der Atmosphäre,Johannes Gutenberg-Universität Mainz, Mainz, Germany.
[5]Max-Planck-Institut for Meteorology, Hamburg, Germany.
[6]Laboratoire d'Optique Atmosphérique, UMR 8518 CNRS/Université des Sciences et Technologies de Lille, Villeneuve d'Ascq, France.
[7]Max Planck Institute for Chemistry, Particle Chemistry Department, Mainz, Germany.
[8]University of Vienna, Faculty of Physics, Aerosol Physics and Environmental Physics, Vienna, Austria.
[*]Now at Flugexperimente, Deutsches Zentrum für Luft- und Raumfahrt (DLR), Oberpfaffenhofen, Germany.
[**]Now at The University of Manchester, Centre for Atmospheric Science, Manchester, UK.

*Correspondence to*: Valery Shcherbakov (v.shcherbakov@opgc.univ-bpclermont.fr)

**Abstract.** Aerosols of the volcanic degassing plumes from Mt. Etna and Mt. Stromboli were probed with *in situ* instruments on board the Deutsches Zentrum für Luft- und Raumfahrt research aircraft Falcon during the contrail, volcano, and cirrus experiment CONCERT in September 2011. Aerosol properties were analyzed using angular scattering intensities and particle size distributions simultaneously measured with the Polar Nephelometer and the Forward Scattering Spectrometer probes (FSSP series 100 and 300), respectively. Aerosols of degassing plumes are characterized by low values of the asymmetry parameter (between 0.6 and 0.75); the effective diameter was within the range of 1.5 – 2.8 µm and the maximal diameter was lower than 20 µm. A principal component analysis applied to the Polar Nephelometer data indicates that scattering features of volcanic aerosols of different crater origins are clearly distinctive from angular scattering intensities of cirrus and contrails. Retrievals of aerosol properties revealed that the particles were "optically spherical" and the estimated values of the real part of the refractive index are within the interval from 1.35 to 1.38. The interpretation of these results leads to the conclusion that the degassing-plumes aerosols were porous with air voids. Our estimates suggest that aerosol particles contained about 18 to 35 % of air voids in terms of the total volume.

## 1 Introduction

The impacts of continuously degassing volcanoes on the environment and human health are well recognized (Delmelle, 2003; Mather et al., 2004). Fumarolic plumes of passively degassing volcanos are strongly involved in the deposition and redistribution of metals and trace elements (Fulignati et al., 2006). Degassing volcanoes have a potentially large effect on the natural background aerosol loading and the radiation budget of the atmosphere (Schmidt et al., 2012; Oppenheimer et al., 2011).





The size-resolved chemical composition, morphology and particle size distribution of volcanic aerosols are important in defining their effects on the atmosphere, environment and human health. A description of the various techniques used in the characterization of 'near-source' volcanic particles as well as a review of field-campaigns results are given by Mather et al., (2004). The morphology of atmospheric

and volcanic-plumes aerosols was studied mostly by electron microscopy and x-rays spectroscopy (see, e.g., Pósfai et al., 1999; Obenholzner et al., 2003; Mather et al., 2004; Martin et al., 2008). Due to the high spatial resolution of such instruments, it was evidenced that complex composition, irregular shape, intricate internal structure, and random surface roughness are the rule rather than the exception for aerosol particles. At the same time, the quasi-spherical particles are not uncommon (see, e.g.,

Obenholzner et al., 2003).

The overwhelming majority of works dealing with internal structure of aerosol particles consider a mixture of chemical compounds (see, e.g., Nousiainen, 2009; Ulanowski and Schnaiter, 2011; and references therein). To date, literature on porous aerosol is rather limited. Jeong and Nousiainen (2014) demonstrated, using transmission electron microscopy analysis, that the internal structures of individual

Asian dust particles were formed by the patterned arrangement of nano-to-micron-sized mineral grains and pores. Highly-porous aerosol particles, obtained from proxies for organic compounds, were investigated by Adler et al. (2013, 2014) with focused ion beam - scanning electron microscopy techniques and a cavity ring down system. Most of the information on volcanic vesicular, i.e., porous ash particles was obtained from samples collected at the ground level. A compilation of the

morphological features of volcanic ash particles derived through the use of scanning electron microscopy was done by Heiken and Wohletz (1985).

Generally, scattering matrices of irregular or/and heterogeneous particles have to be modelled on the base of the exact electromagnetic wave theory (see, e.g., Mishchenko et al., 2000). On the other hand, heterogeneous materials could be treated similar to homogeneous substances when the typical

dimension $d$ of their inhomogeneity is much smaller than the wavelength $\lambda$ of the considered radiation, $d<<\lambda$ (Chýlek et al., 2000), that is, using effective medium approximations (EMAs). The applicability of EMAs for calculating scattering properties of inhomogeneous atmospheric particles was verified in the recent work by Liu et al. (2014) by comparing with the standards-of-truth provided by calculations based on the pseudo-spectral time domain method (PSTD). The exact heterogeneous mixing structures

of non-absorptive or weakly absorptive particles were considered in the PSTD calculations. Liu et al. (2014) concluded that scattering properties of the equivalent homogeneous particles agree well with those of internally mixed particles.

Of course, it is hardly probable that EMAs are able to provide all elements of a scattering matrix with high accuracy for any value of the scattering angle $\theta$. But, it is feasible to perform modeling with

admissible errors, especially when one deals with angular scattering intensities within a limited interval



of $\theta$. For example, the works by Chylek et al. (1988) and Kolokolova and Gustafsonm (2001) showed acceptable agreement between experimental results and EMA calculations.

Remote sensing methods, namely, sun photometry, lidar sounding and satellite observations are largely employed for volcanic-plumes studies. Generally, such techniques are related to inverse problems, i.e.,

retrievals of aerosols characteristics on the base of measured optical signals. Consequently, values of the aerosol refractive index have to be estimated with a retrieval code or specified a priori, i.e., preassigned. When values are preassigned, the refractive index of bulk matter or sulfuric acid solutions are commonly used (see, e.g., Watson and Oppenheimer, 2000, 2001; Spinetti et al., 2007; Kahn et al., 2007; Martin et al., 2009; Marenco et al., 2011; Young et al., 2012). Examples, where the refractive

index was retrieved along with a size distribution, can be found in the works by Toledano et al. (2012); Waquet et al., (2014), and references therein. Effective medium approximations were used in inverse problems by Abo Riziq et al. (2007) and Adler et al. (2013, 2014) where refractive indices of laboratory-generated aerosols were retrieved by comparing the measured size-resolved extinction-efficiency with Mie-theory calculations.

This work is devoted to aerosol optical characteristics of the volcanoes Mt. Etna and Mt. Stromboli degassing plumes, retrievals of size distributions and refractive-index values, and consequent inferences about morphological properties of aerosols. In the following, the aircraft instrumentation and the measurements conditions are outlined first. Section 3 addresses the outcomes of the Principal Component Analysis in details. Section 4 describes a careful approach to derive size-distributions and

refractive-index as well as the results. Section 5 is devoted to discussion and inferences.

## 2 Instrumentation and flight overview

This study addresses *in situ* measurements in degassing plumes from the volcanoes Mt. Etna and Mt. Stromboli performed during the CONCERT (contrail, volcano and cirrus experiment) campaign in 2011 (Voigt et al., 2014a). Particles and trace gases of the volcanic plumes were probed with instruments on

board the DLR (Deutsches Zentrum für Luft- und Raumfahrt) research aircraft Falcon. Descriptions of the instrumentation of the two CONCERT campaigns in 2008 and 2011 are given in Voigt et al. (2010, 2011, 2014a); individual instruments are discussed in detail by Frey et al., (2011); Gayet et al., (2012); Jeßberger et al., (2013); Kaufmann et al., (2014); Kaufmann et al., (2016) and Jurkat et al., (2015). In addition to the instrumentation of the CONCERT campaign in 2008 (Voigt et al., 2010), the FSSP-100

forward scattering spectrometer probe was employed during the CONCERT campaign in 2011. Below, we briefly describe the instruments used for our study.

### 2.1 Particle probes

The Polar Nephelometer (PN) (Gayet et al., 1997) measures the angular scattering intensities (ASIs) of an ensemble of particles ranging from less than 1 micrometer to about 1 mm diameter, which intersect a



collimated laser beam ($\lambda$ = 804 nm) near the focal point of a parabolic mirror. Observations are usually limited to 32 scattering angles $\theta$ near-uniformly positioned from 15° to 162°. Measurements at nearly forward and backward directions ($\theta < 15°$ and $\theta > 162°$) are not reliable due to the diffracted light pollution caused by the edges of holes drilled on the paraboloidal mirror (see, e.g, Jourdan et al., 2010).

The sampling volume is defined by the cross sectional area of the beam (10 mm long and 5 mm diameter) multiplied by a linear speed of air populated with particles passing through the instrument. Direct measurement of the ASIs allows particle types (droplets or aspherical particles) to be distinguished and calculation of the optical parameters to be performed (extinction coefficient *Ext* and asymmetry parameter *g*, see Gayet et al., 2002). Generally, the *g*-value decreases with increasing

asphericity of the particles (Gayet et al., 2002; Gayet et al., 2012). The accuracies of the extinction coefficient and asymmetry parameter are 25% and 4% respectively (Jourdan et al., 2010; Gayet et al., 2012).

The FSSP-300 and FSSP-100 forward scattering spectrometer probes (Baumgardner et al., 1992; Petzold et al., 1997; Frey et al., 2011) measure the intensity of light scattered by single particles in

forward direction at angles from 6° to 15° (Jeßberger et al., 2013), which is governed mainly by diffraction and therefore depends on (i) the refractive index of the particles, and (ii) the projected area of the particle which itself depends on the particle shape. Indeed, the size calibration for aspherical particles is expressed in terms of equivalent surface diameter, i.e. the diameter of a sphere that has the same projected area (Mishchenko et al., 1997). The FSSP-300 signal is resolved into an array of 31

channels in the size range of 0.35 to 38.6 µm. The signal is then converted into a corresponding particle size using water droplet response (the refractive index of 1.33). Because of the ambiguities in the Mie scattering curve, the FSSP-100 size-distributions have been rebinned to 14 size bins in the diameter range from 1.02 to 47.05 µm (water droplet).

## 2.2 Trace-gas instruments

The trace gases $SO_2$, HCl, and $HNO_3$ were measured by the Airborne chemical Ionization Mass Spectrometer AIMS (Voigt et al., 2014a; Jurkat et al., 2015) equipped with an inlet system and a custom made ion source generating $SF_5^-$ reagent ions (Jurkat et al., 2010). Air is drawn into the backward-facing perfluoroalkoxy (PFA) inlet using a pumping system. In the high-voltage gas discharge ion source $SF_5^-$ ions are produced from a flow of $SCF_8$ in $N_2$ at pressures controlled to 40 hPa. The $SF_5^-$ ions react

selectively with trace species to form product ions via a fluoride transfer reaction (Jurkat et al., 2011). The ions pass three differentially pumped vacuum chambers of the mass spectrometer system, before they are detected with a channeltron detector. An in-flight calibration is performed by adding a permanent flux of isotopically labeled $^{34}SO_2$ to the inlet line. Further, $HNO_3$ and HCl were calibrated in the laboratory after the flights. Accuracies of 17–36% were achieved for $SO_2$, 33–35% for HCl, and 18–

36% for $HNO_3$, mainly depending on dilution of the inlet flow (Voigt et al., 2014a; Voigt et al., 2014b).





Water vapor was measured with the tunable diode laser system WARAN (WAter vapoR ANalyzer), which primarily consists of a commercial WVSS-II water vapor sensor (SpectraSensors Inc.) (Kaufmann et al., 2014) with the measurement range from 30 to 40000 ppmv. In the measurement cell, a laser is tuned over the water vapor absorption line at 1.37 μm and the water vapor mixing ratio is determined from the absorption signal. The passive sample flow through the system is realized using a Rosemount inlet. The water vapor mixing ratio is measured with a relative uncertainty < 6 % at 2.4 s time resolution and for water vapor mixing ratios above 500 ppmv relevant for the low-altitude flight legs shown here. The instrument is calibrated before and after the campaign using a frost point hygrometer (MBW373LX) as the reference. The uncertainty of 0.5°K in the static air temperature measurement translates to a relative uncertainty of around 4% in the saturation pressure over water. Adding both contributions, the relative humidity with respect to water ($RH_w$) can be determined with a relative uncertainty of a $RH_w$ value generally less than 10 %.

### 2.3 Flight overview

Degassing plumes from the volcanoes Mt. Stromboli and Mt. Etna were probed on 30 September 2011 (from 6:50 to 8:40 UTC). Visual observations from the cockpit of the Falcon showed that there was one single plume degassing from the Mt. Stromboli; whereas there were two distinct degassing plumes from the Mt. Etna, see Fig. 1(a). The Etna plumes were spreading at two different altitudes and they originated from at least two craters, namely, North East crater (upper plume) and Bocca Nuova crater (lower plume) (Voigt et al., 2014a). From the four main craters at Mt. Etna, these two craters were identified as the major emitters on that day. The Falcon flight track over Sicily is shown in Fig. 1(b). It was set up to probe the two Etna plumes and the Stromboli plume.

Figure 2 shows time series of trace-gases measurements, flight altitude, extinction coefficient and asymmetry parameter both derived from the Polar Nephelometer measurements. The very low relative humidity has to be particularly emphasized. The values of $RH_w$ are lower than the crystallization humidity of most sulfates, nitrates and chlorides (see, e.g., Tang and Munkelwitz, 1991). In other words, the plumes were spreading in the dry troposphere. In addition, the air temperatures within the plumes were always above the freezing point, i.e., > 273.15°K. The extinction-coefficient series reveals several peaks that correspond to the time intervals when the research aircraft was probing the degassing plumes (see shaded areas in Fig. 2). The close correspondence between the extinction coefficient and the time series of the trace gases $HNO_3$, $SO_2$, $HCl$ is evident in Fig. 2. The trace-gas composition of the volcanic plumes has been investigated by Voigt et al., (2014a). Different trace gas mixing ratios of $CO_2/SO_2$ and $SO_2/HCl$ has been identified in the volcanic plumes from North East and Bocca Nuova craters. Specifically, it is shown that neither $SO_2$ conversion to sulfate nor $HCl$ uptake in sulfate aerosol play a major role in the aging plume under dry tropospheric conditions.





## 3 PCA analysis of Polar-Nephelometer data

In this section, a statistical tool, the Principal Component Analysis (PCA) is applied to the Polar Nephelometer data obtained during the CONCERT campaign in 2011. It should be emphasized that the peculiarity of the PN data recorded within the quiescent degassing volcanic plumes is that the signals

were quite low, at least much lower than those ones registered within contrails and clouds. We recall that during particle samplings, the Polar Nephelometer raw signals are superimposed on the background signals (or zero baselines), which are due to electronic or optical noises. As a consequence, the accurate subtraction of the background signals was of utmost importance (see Shcherbakov et al., 2006). Thus, the corresponding data were preprocessed with special-purpose software and operator supervision of the

data-treatment quality, i.e., wavelet denoising technique.

### 3.1 Clustering techniques

We employed the PCA to carry out the cluster analysis (see, e.g., Jolliffe, 2002, Ch. 9; Jourdan et al. 2010), that is, to separate the recorded angular scattering intensities (ASIs) into clusters. The advantage of the PCA consists in the fact that it is self-sufficient; in other words, no a priori hypotheses are

15 needed, the clustering is only based on internal properties of a data set. In brief, the PCA is an unsupervised method used to explore the intrinsic variability of the data.

The PCA is particularly fruitful when one deals with high-dimensional data. The PCA provides the possibility to capture much of the total data variation in a few dimensions and to organize observed data into meaningful clusters. That is, with PCA technique, each vector of the data can be represented

adequately by a few coefficients, usually three, which correspond to eigenvectors of a covariation matrix. This is especially true when a limited number of primary physical parameters have a major impact on the measured functions (in our case, angular scattering intensities). The obtained coefficients can be used in visualization of the data by scatter plots.

Algebraically, principal components could be defined as particular linear combinations of a set of

25 variables. These linear combinations represent the selection of a new coordinate system obtained by rotating the original system of coordinates. The new axes correspond to the directions with maximum variability and provide a simpler description of the covariance structure of the original set of variables (Johnson and Wichern, 1998).

The peculiarity of our approach is that the log-transformation is applied to the angular scattering

intensities before the PCA is carried out (see details in Jourdan et al., 2003; Shcherbakov et al., 2005). The reason for such a preprocessing is the following. In contrast to phase functions, ASIs are not normalized, e.g., they are proportional to the particle concentration. In addition, the ASIs values, as well as the variances, at the forward angles, as a rule, are of orders of magnitude higher than that ones at the sideward and backward angles. In such conditions, the conventional PCA yields the first principal

components that only follow the variance at the forward angles; they do not reproduce all the ASIs





variability. The preprocessing, that is, the logarithm of ASIs values makes the variances to be comparable at all angles. In the strict sense, the log-transformation leads to the multiplicative model of the variance analysis, in contrast to the conventional PCA, which is based on the additive model. At the same time, the variance can be discussed in terms of the additive model when ASIs are plotted in the

log-scale.

Expressed mathematically, our approach leads to the following representation of measured angular scattering intensities $\sigma_j(\theta_i)$ in terms of the principal components $\xi_l(\theta_i)$, that is, the first $k$ eigenvectors of the correlation matrix of the log-transformed data-set:

$$\ln[\sigma_j(\theta_i)] \approx \langle \ln[\sigma(\theta_i)] \rangle + \sum_{l=1}^{k} C_{j,l}\xi_l(\theta_i),$$

where $\theta_i$ is the scattering angle, the index $i$ designates the $i$-th scattering angle and takes values from 1 to 32, the index $j$ refers to the $j$-th observation of the analyzed flight sequences, $<...>$ denotes averaging over the total data set (i.e., 1803 PN measurements). When vector expressions are employed, $\vec{\sigma_j}$ has the components $\sigma_j(\theta_i)$; $\vec{\xi_l}$, $\overrightarrow{\ln\sigma_j}$, and $\overrightarrow{\ln\sigma}$ have the components $\xi_l(\theta_i)$, $\ln[\sigma_j(\theta_i)]$, and $\langle \ln[\sigma(\theta_i)] \rangle$, respectively. The coefficients $C_{j,l}$ are computed as: $C_{j,l} = \left(\overrightarrow{\ln\sigma_j} - \langle\overrightarrow{\ln\sigma}\rangle\right)^T \cdot \vec{\xi_l}$, where $^T$ denotes a

transposed matrix. Each vector (i.e., $\overrightarrow{\ln\sigma_j}$ at each $j$) is represented by a few coefficients $C_{j,l}$ with quite good accuracy.

### 3.2 Clustering results

The PCA results were obtained using the CONCERT data recorded during the flights on 16 September 2011 (from 14:40 to 17:50 UTC) (Kaufmann et al., 2014) and on 30 September 2011 (from 6:50 to 8:40

UTC) (Voigt et al., 2014a). The data set contains the ASIs measured in degassing plumes, cirrus and contrails. Such a large set was chosen to point out that the scattering pattern of degassing plumes is clearly distinctive (see below).

When the PCA is applied to ASIs, the principal components can have a clear physical meaning. Figure 3(a) shows the first three principal components along with the corresponding eigenvalues normalized as

a percentage of the total variance. The first vector $\vec{\xi_1}$ almost does not depend on the scattering angle, it accounts for 96.8 % of the data variability. This means that it is closely linked to the extinction coefficient and about 97 % of the ASIs variation is mostly due to fluctuations of the particles concentration.

It is well known that effects of a size distribution, refractive index, particles shape, surface roughness,

are better represented using phase functions, i.e., normalized ASIs. Thus, not only the other principal components are of importance, their contribution to the remaining variability has to be evaluated.

The second vector $\vec{\xi_2}$ represents 70.4% of the remaining variability; it has negative values at sideward angles. Generally, high negative values of the corresponding coefficients $C_{j,2}$ of such a vector take into





account the irregular shape and/or the deep surface roughness of large particles. And, all of this leads to lower values of the asymmetry parameter (see, e.g., Jourdan et al., 2010).

The third vector $\vec{\xi_3}$ represents 15.6% of the remaining variability. Its shape reveals that $\vec{\xi_3}$ is related to the forward/backward-hemisphere partitioning of the scattering. The high negative values of the corresponding coefficients $C_{j,3}$ imply that less energy is scattered in the forward hemisphere and, thus, more energy is scattered in the backward hemisphere.

Figure 3(b) shows the scatterplot of the $C_{j,3}$ expansion coefficient versus the $C_{j,2}$ coefficient. Therefore, such a presentation describes the features of the PN data set in one of the clearest and informative ways. Each point is directly associated with one of the measured ASCs. Three main distinctive clusters of points can be identified in Fig. 3(b). Additional analysis of the total CONCERT-2011 data set provided possibility to associate the clusters with the measurements performed within cirrus clouds, contrails and the degassing volcanic plumes (see the notations in Figure 3(b)).

The negative $C_{j,2}$ values of the cirrus cluster are typical of irregular large ice particles (see, e.g., Jourdan et al., 2010). The contrails cluster is characterized by high positive values of $C_{j,2}$. This feature means that the corresponding phase functions are quite similar to those of ensembles of large spherical particles. The fraction of spherical particles increases with increasing $C_{j,2}$. This is especially true for the young-contrails sub-cluster. A detailed discussion can be found in Gayet et al. (2012).

The peculiarity of the degassing-plumes cluster is that the coefficients $C_{j,2}$ are close to zero. Thus, the properties of the degassing-plumes ASIs can be discussed just in terms of the vector $\vec{\xi_3}$. That fact results in the following. The effective diameter and the asymmetry parameter of the degassing-plumes particles have to be smaller, if not much smaller, compared to those ones of other clusters of the data set, among them the young contrails. Another property of the degassing-plumes cluster is that it falls into sub-clusters. Figure 3(c) repeats Fig. 3(b) but with the focus of attention on the degassing plumes. It is seen that the ASIs of the NECa and NECb plumes are close to each other; they form one cluster. The BNa and BNb clusters are a little bit separated with the third coefficients, but belong to the same range of the second coefficient that accounts for 70.4% of the ASIs shape variability. Other properties of the degassing-plumes sub-clusters will be discussed in details below.

## 4 Number concentrations, effective radii and refractive indices of the volcanic plume layers

In this work the intervals of the degassing-plume penetration were specified on the basis of the PN data after subtraction of the background signals, which correspond to the electronic noise and ASIs of the free atmosphere. The angular scattering intensities (ASI) from the Polar Nephelometer were averaged over the penetration intervals, i.e., over 17 seconds for Stromboli (SV) and over 143, 172, 122, 80 and 313 seconds for Etna's Bocca Nuova (BNa and BNb) and North East (NECa1, NECa2, and NECb)





crater plumes, respectively (see notations in Fig. 1). The averaged ASIs were used to retrieve microphysical and optical characteristics of degassing-plumes aerosols.

### 4.1 Retrieval techniques and software

Our retrievals were performed with the software and the pre-calculated kernels developed by Dubovik
et al. (2006). We recall that the software is employed in the operational processing of AERONET (AErosol RObotic NETwork) for retrieving detailed properties from observations of ground-based sun/sky-radiometers (Eck et al., 2008). We will briefly describe the main features of the retrieval algorithm and its implementation on our specific dataset.

The pre-calculated kernels contain optical characteristics of spheroids mixtures. The surface of particles
can be smooth or severely rough (Yang et al., 2013). A mixture of spheroids of different aspect ratios, sizes and surface texture (smooth or severely rough) is employed as a generalized aerosol model (representing spherical, aspherical, and mixed aerosols).

The code offers the possibility to retrieve a complete set of aerosol parameters, including the complex refractive index and the size distribution. As for size-distribution retrievals, the undoubted advantage of
the Dubovik code consists in the fact that solutions are constrained to be non-negative, which significantly improves the quality of retrievals. The constraint is imposed through an elegant and well-founded approach, more specifically, the assumption of the log-normal distribution of measurement errors (see, e.g., Dubovik et al., 2011).

The code includes a quite large set of input parameters that is very advantageous for an experienced
user. For example, different hypotheses of aerosol composition, among others, effects of the ultrafine fraction, can be tested. On the one hand, supervised retrievals are time consuming; on the other, one has the possibility to assure a high quality of retrievals. We list some criteria of the retrieval quality in the following. Final results are scarcely affected by small variations of input parameters. When the regularization-parameter value is chosen according to the "L-curve" method (see, e.g., Hansen, 1992),
the retrieval residuals correspond to the measurement errors. In a generally non-linear case, a particular attention must be given to the verification whether the obtained solution corresponds to the global minimum of an objective function. As for aerosol-characteristics retrievals, different starting values of the refractive index may be randomly tried.

Summarized, we modeled the degassing plumes as a mixture of two main particle fractions; one
consists of spherical and one of aspherical particles. Although aerosol particles were not ellipsoidal, we consider randomly oriented spheroids as a reasonable approximation of an ensemble of quite small aspherical aerosols. In addition, we recall that we considered smooth and severely rough particles (Yang et al., 2013).



## 4.2 Retrieval results

Three representative examples of retrievals are shown in Fig. 4. The corresponding time-sequences SV, BNa and NECb are identified in the time-series in Fig. 2 by shadowed areas. In Fig. 4 (a-c), the left panels display the particle size distributions measured with the FSSP-300 and FSSP-100 (solid lines) and the retrieved one (solid black circles). It should be pointed out that the FSSP measurements in Fig. 4 represent particle size distributions considering the size response to water droplets. Due to size response uncertainties linked to different refractive indexes and/or particle shape in plume aerosols (Pinnick and Auvermann, 1979; Dye and Baumgardner, 1984; Jaenicke and Hanusch, 1993; Febvre et al., 2012), the comparison with retrieved size distributions should only be considered as qualitative in Fig. 4. Despite these limitations, the FSSP data and the retrieved distributions reveal very similar features of the degassing-plumes aerosols, in particular the absence of large particles. The calibrated probe response of the specific FSSP adjusted to different refractive indexes is beyond the scope of this paper.

The size distribution was retrieved along with the refractive index $m = n + i \chi$ and the spherical/aspherical partitioning ratio (SAR) of aerosol particles (see Table 1). In the following, the SAR defines the percentage in number of spherical particles relative to the total number of particles. Furthermore, the refractive index and the partitioning ratio are assumed to be constant over the full size range of the retrieved particle size distribution. We emphasize that the information content of the PN data is inadequate to retrieve size-dependent values of such parameters. In other words, variations of the shape and/or the refractive index of small particles lead to variations of the phase function that are lower than the PN measurement errors. Therefore, the refractive index and the partitioning ratio are assumed to be constant over the full size range.

The right panels of Fig. 4 (a-c) represent the following data. The average scattering phase function (without normalization) measured by the Polar Nephelometer is shown by solid red circles (see the averaging time intervals in Table 1). The retrieved phase function, shown by solid black circles, was computed from the retrieved size distribution. The main result of Fig. 4 is that the retrieved phase functions agree well with the observations. And, it should be underscored that there is no systematic bias between them. The residuals *Res* were computed with the formula

$$Res = 100 \cdot \sqrt{\frac{1}{N} \sum_{i=1}^{N} \left[ \frac{I(\theta_i)_{ret} - I(\theta_i)_{meas}}{I(\theta_i)_{meas}} \right]^2},$$

where $I(\theta_i)$ is the light intensity at the scattering angle $\theta_i$, and the subscripts "*meas*" and "*ret*" refer to the measured and retrieved values, respectively. The low values of the residuals are noteworthy as well, see Table 1.

Table 1 summarizes the measured and retrieved characteristics. As mentioned above, the relative humidity $RH_w$ was low, i.e., about 36 % within the Stromboli plume and near 10 % during the Etna



plume samplings. The air temperature was positive, including at the Etna upper-plume altitude (0.3°C). The number concentration (*Conc*) of aerosols with the diameter $d > 0.9$ μm is estimated within the range of $25 - 51$ cm$^{-3}$. The effective diameter (*Deff*) was low, between 1.5 and 2.8 μm, where both parameters *Conc* and *Deff* were deduced from the FSSP data. The retrieved values of the refractive

index belong to the interval from 1.35 to 1.38. Low values [0.65 – 0.70] of the asymmetry parameter *g* correspond to ensembles of small particles, which is in agreement with the *Deff* values. The extinction coefficients *Ext* were computed form the retrievals data, the values are quite small confirming that the ASIs were recorded in single-scattering conditions. We recall that the optical characteristics correspond to the wavelength of 0.8 μm and that the all parameters were recorded at distances more than 6 km from

the plumes sources.

As it was mentioned above, the ASIs of the degassing plumes are partitioned into sub-clusters (see Fig. 3c). The ASIs of the NECa and NECb plumes are spread within a cluster that is well distinguished from other ones. The second coefficients $C_{j,2}$ of the BNa and BNb data are within the same range of values. (We recall that the second vector $\vec{\xi_2}$ represents 70.4% of the ASIs shape variability.) Consequently, the

asymmetry-parameter values of the Bocca Nuova plumes (BNa and BNb) are a little bit higher compared to those ones of the North East crater (NECa1, NECa2, NECb), see Table 1. The histograms in Fig. 5 show probability distribution functions of the asymmetry-parameter values computed from the measured ASIs of the BNa and NECb plumes. These two distributions were selected only by reason of larger data sets, i.e., higher statistical significance. Generally speaking, the BNb and NECa distributions

are close to histograms of the BNa and NECb, respectively. As it is seen on Fig. 5, the distributions are narrow suggesting quite homogeneous aerosol-plumes properties. The distinction between optical characteristics of the BN and NEC plumes could result from the difference in chemical composition of aerosols (Voigt et al., 2014a). Though completely independent, this classification of the individual volcanic plumes is similar to results from trace gas composition measurements of Voigt et al., (2014a)

for plumes SV, BNb, NECa and NECb. In addition, the difference in the BN and NEC aerosol emissions was underscored earlier by Allen et al. (2006), who studied aerosol particles at the summit of Mt. Etna downwind from the degassing vents, and by Scollo et al. (2012), who reported observations of the Multi-angle Imaging SpectroRadiometer. Martin et al., (2008) evidenced persistent differences in the size distributions of sulfate aerosols between the two main Etna summit plumes.

**5 Discussion and inferences**

Our results in Table 1 show that the spherical/aspherical partitioning ratio (SAR) value is of 100% for the all considered cases of degassing plumes. In other words, the best fits of the Polar Nephelometer data were obtained with the model of spherical aerosol-particles. It must be emphasized that this result does not mean that aerosols were perfect spheres. Since (i) the aerosol-particles were quite small with



respect to the wavelength of 0.8 μm (see the effective-diameter values in Table 1), (ii) the angular scattering intensities (ASIs) were measured within the limited range of angles for the relatively small number of $\theta_i$, (iii) the ASIs were averaged over an ensemble of particles, and (iv) the PN data were affected by measurement errors, we only can conclude that the aerosols of the degassing plumes were

5 "optically spherical" (see, e.g., Dick et al., 1998). The term "optically spherical" should be considered within the context of optical instrumentation. For example, the model of spherical particles matches well the PN experimental data; and we believe that the asymmetry-parameter estimates are trustworthy. At the same time, it might be that polarization measurements, particularly in the backward hemisphere, will reveal nonspherical features of degassing-plumes aerosols.

The retrievals indicate that the aerosol particles were either non-absorbing or weakly absorbing with an upper bound for the imaginary part of the refractive index of $10^{-4}$. Variations of the phase functions for the imaginary part within the interval $[0; 10^{-4}]$ are smaller than the measurement errors (Verhaege et al., 2008).

The most important and somewhat unexpected result of this work is the fact that the retrieved values of

15 the real part $n$ of the refractive index belong to the interval from 1.35 to 1.38 (Table 1).

Degassing-plumes aerosols of Mt. Etna were probed at distances greater than 6 km from the sources, i.e., the volcanic craters. The recorded values of the horizontal air-speed at the flight altitudes were 5.9 m/s or lower. Consequently, our measurements are related to aerosols that were generated more than 16.5 minutes before. Large aerosols tend to settle quickly out of the atmosphere. High initial

concentrations of fine particles decay rapidly in the atmosphere due to coagulation and dilution, so that measurements depend on the distance from the source and the wind conditions (Ammann and Burtscher, 1990). Thus, it is hardly possible to directly compare our *in situ* data and characteristics of particles sampled on filters at the ground. Nevertheless, some ideas on the chemical composition of degassing-plumes aerosols can be drawn from published results.

Electron-microscopy analysis of particles in the range of diameters 5 − 100 nm showed significant levels of silicate nanoparticles in the Mt. Etna plumes (Ammann and Burtscher, 1990; Martin et al., 2008). As it was mentioned above, the Dubovik code provides possibility to evaluate effects of the ultrafine fraction, i.e., nanoparticles, on retrieval results. Our simulations and tests led us to the following conclusions. The information content of the PN data is inadequate for a correct estimation of

the ultrafine-fraction number-concentration. Variations of that number concentration within a large range of values do not affect the retrieved values of the refractive index.

According to chromatographic analysis of ionic species of soluble particles sampled on filters from degassing plumes near the crater rims of Mt. Etna (Allen et al., 2006; Martin et al., 2008), aerosols are mainly composed of sulfates. Fluorides, chlorides and nitrates are present as well. Besides, silicates

were observed in Etna degassing plumes (Lefevre et al., 1986; Martin et al., 2008). Generally, the values of the bulk refractive index at $\lambda=0.8$ μm of sulfates and other soluble inorganic species presumed





to form the degassing plumes belong to the interval 1.48 – 1.58 or higher (see, e.g., Toon et al., 1976; Tang, 1996; Lide, 2010). The exception is some alkali halides with $n$ about 1.40 or lower. For example, the refractive index values at $\lambda$=0.8 μm of NaF, KF, and MgF2 are 1.323, 1.36, and 1.375, respectively (Li, 1976). As for silicate particles, they have quite large values of the bulk refractive index, e.g., $n$ is

5 about 1.46 for amorphous silica glass (see, e.g., Kitamura et al., 2007), 1.52 – 1.59 or higher for feldspars (Lide, 2010), and about 1.56 – 1.99 or higher for garnets and other silicate minerals (Lide, 2010).

It is obvious that the retrieved values $n$ (Table 1) are substantially lower than the bulk refractive index of sulfates and other inorganic matter presumed to form the degassing plumes. That result was

10 thoroughly verified. The Dubovik code provides possibility to hold fixed a refractive-index value $n_{fix}$ while a size distribution is retrieved, and to compute scattering characteristics for given optical constants and the retrieved size distribution. As it is expected, the residuals $Res$ increased when the fixed value $n_{fix}$ of the refractive index deviated from the retrieved one. Two important points are to be underscored. When $n_{fix}$ was higher than 1.40, (i) the $Res$ values significantly exceeded the

15 measurement-errors level; (ii) the plots of the reconstructed angular scattering coefficients, that is, computed for $n_{fix}$ and the corresponding size distribution, clearly manifested a systematic deviation from the plots of the measured ASIs. An illustration of the systematic deviation is given in Fig. 6. The phase functions, i.e., the normalized ASIs, were computed for the retrieved size distribution of the NECb case (see Fig. 4) and a set of values of the real part $n$ of the refractive index. It is seen that the

20 behaviour of the curves within the scattering-angles interval of 15 – 150 degrees is sensitive to the $n$ value. Not only the slopes of the curves are different but the local minimum shifts monotonically from 114° ($n$=1.33) to 124° ($n$=1.58). The gray bar in Fig. 6 indicates the range of the PN operating angles. Thus, we can say that the accuracy of the refractive-index estimation is based on the functional behaviour of measured ASIs.

It is unlikely that the relative concentration of alkali halides increased at distances greater than 6 km from the sources so that their optical properties became dominant. And in our particular case, the hypothesis of aqueous solutions or suspensions in the Mt. Etna plumes has to be rejected as well because of the very low relative humidity at the flight altitudes (Table 1). The measured $RH_w$ values (about 10%) are much lower than the efflorescence $RH_w$ of the overwhelming majority of the mentioned

above chemical species (see, e.g., Table 3, Martin, 2000). We recall that efflorescence is a specific process in the more encompassing concept of crystallization, it specifically involves water vapor (Martin, 2000). In other words, the conditions were favorable even for homogeneous nucleation. As for the chemical systems NaNO3/H2O, NH4HSO4/H2O, and NH4NO3/H2O that do not readily crystallize at the lowest $RH_w$ values (Martin, 2000), the conditions were favorable for heterogeneous nucleation due

to significant levels of silicate nanoparticles in the Mt. Etna plumes (Martin et al., 2008). The hypothesis of unsteady state conditions was discarded in view of the data of the PCA analysis. For





example, NECb plume was probed at distances from the craters between 5.9 and 47.7 km, respectively. Such distances correspond to the aerosol age between 16.5 minutes and two hours. There is no trend in *g* values of the NECa plume. A slight rise at the end of the NECb *g* series may be due to increased measurement errors at low aerosol concentrations.

In view of the results above, the most reasonable conclusion to make is that the aerosol particles of the degassing plumes were porous with air voids. Therefore, the retrieved values *n* (Table 1) correspond to the refractive index of the effective medium instead of the bulk one.

That conclusion is in agreement with data of transmission electron microscopy of aerosol particles collected in the remote marine troposphere (Pósfai et al., 1999). Figure 3(a) by Pósfai et al. (1999) is
especially noteworthy. It shows a quasi-spherical ammonium-sulfate aerosol particle of about 1 μm diameter. The particle has the onion-like structure with soot inclusions. Obenholzner et al., (2003) investigated the micro-morphology of aerosol particles from the passively degassing plume of Popocatepetl volcano using the Field emission scanning electron microscopy (FESEM). A large set of spherical particles were observed; including spongy, i.e., porous aerosols and spheres, enclosing a
dozen small crystals Obenholzner et al., (2003).

Microwave analog experiments performed by Kolokolova and Gustafson, (2001) corroborate our conclusion as well. The measured ASIs of inhomogeneous particles show reasonable agreement with effective medium approximations (Kolokolova and Gustafson, 2001). Moreover, porosity could be the cause of the quite low values of the refractive index of the Eyjafjallajökull volcanic aerosol retrieved
from POLDER/PARASOL measurements (Waquet et al., 2014).

At the wavelength of 0.8 μm, the real part of the bulk refractive index of sulfates, nitrates and other inorganic matter presumed to form the degassing plumes belongs to the range about 1.48 – 1.58. Considering that range, we employed the Maxwell Garnett mixing rule (Maxwell Garnett, 1904; see also Kolokolova and Gustafson, 2001) to estimate the volume fraction *f* of the inorganic matter that
leads to the refractive index of the mixture about [1.35 – 1.38]. In other words, the inorganic matter and the air voids were taken as the matrix and the inclusions, respectively. We have obtained the interval of [0.65 – 0.82] as the estimate for the volume-fraction value. This means that aerosol particles of the degassing plumes contained 18 – 35 % of air voids (in terms of the total volume). The estimated volume fraction *f* and the bulk-refractive-index range of inorganic matter would lead to the refractive index of
aerosols about [1.43 – 1.53] if pores were filled up with water. The last assessment is in agreement with the results by Waquet et al., (2014).

Our finding has the following consequences. (i) Climate models, optical methods of remote sensing and optical instruments of particles counting have to consider not the bulk- but the effective refractive index of volcanic particles and aerosols in general. (ii) The volume fraction *f* of bulk matter has to be taken
into account in assessing the volcanic ash spreading and loading. (iii) The volume fraction *f* has to be





considered in aerosols sizing instrumentation based on the inertia effect. (iv) Aerosols porosity could be of importance for condensation and freezing processes.

## 6 Conclusions

Volcanic degassing plumes were probed on 30 September 2011 with *in situ* instruments onboard the
DLR Falcon research aircraft during the CONCERT experiment. The plumes were spreading in the dry troposphere at temperatures above the freezing point.

Aerosols of degassing plumes from the volcanoes Mt. Etna and Mt. Stromboli are characterized by quite low values of the asymmetry parameter (between 0.6 and 0.75); their scattering features are clearly distinctive from angular scattering intensities of cirrus and contrails.

The measured and the retrieved size distributions of the degassing-plumes aerosols are in good agreement. The effective diameter of the aerosols was within the range of 1.5 – 2.8 µm, the maximal diameter of particles was lower than 20 µm.

Retrievals of aerosol properties revealed that the particles were "optically spherical"; the estimated values of the real part of the refractive index belong to the interval from 1.35 to 1.38, which is
substantially lower than the bulk refractive index of sulfates, nitrates and other inorganic matter presumed to form the degassing plumes.

That property leads to the conclusion that the aerosol particles of the degassing plumes were porous with air voids. Our estimates, based on the Maxwell Garnett mixing rule, suggest that aerosol particles of the degassing plumes contained 18 – 35 % of air voids in terms of the total volume.

**Acknowledgments**

We thank the DLR flight department for excellent support during the CONCERT campaign. The campaign was organized by the Research Group AEROTROP under HGF contract VH-NG-309. Part of this work is funded by the Research Cluster VAMOS at Johannes Gutenberg University, Mainz and by the German Science Foundation DFG within SPP1294 HALO. J.-F. Gayet is grateful to DLR for having
provided a guest scientist opportunity at the Institut für Physik der Atmosphäre during this study.

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




**Table 1.** Mean parameter values in the indicated time-intervals (see Fig. 1). The parameters are: time interval, altitude, air temperature $T$, relative humidity with respect to water ($RH_w$); concentration ($Conc$) of particles ($d > 0.9$ μm) and effective diameter $Deff$ measured with the FSSPs; real $n$ and imaginary $\chi$ part of the refractive index, spherical/aspherical partitioning ratio ($SAR$), asymmetry parameter $g$, extinction coefficients ($Ext$), and residual ($Res$) estimated from the retrievals data. (See notations in Fig. 2.)

| | UT (h :min :s) | Alt. (m) | $T$ (°C) | $RH_w$ (%) | $Conc$ (cm⁻³) | $Deff$ (μm) | $n$ | $\chi$ | $SAR$ (%) | $g$ | $Ext$ (km⁻¹) | $Res.$ (%) |
|---|---|---|---|---|---|---|---|---|---|---|---|---|
| SV | 7 :10 :28  7 :10 :45 | 970 | 17.2 | 36 | 32 | 2.8 | 1.35 | 0 | 100 | 0.65 | 0.056 | 6.5 |
| BNa | 7 :29 :10  7 :31 :33 | 3518 | 1.2 | 10 | 51 | 1.9 | 1.36 | 0 | 100 | 0.69 | 0.059 | 7.8 |
| BNb | 8 :22 :45  8 :25 :37 | 3202 | 3.4 | 10 | 25 | 1.5 | 1.37 | 0 | 100 | 0.70 | 0.020 | 9.4 |
| NECa1 | 7 :47 :03  7 :49 :05 | 3656 | | | | | 1.36 | 0 | 100 | 0.66 | 0.059 | 6.7 |
| NECa2 | 7 :51 :10  7 :52 :30 | 3645 | 0.3 | 11 | 40 | 2.2 | 1.38 | 0 | 100 | 0.65 | 0.120 | 7.4 |
| NECb | 8 :30 :13  8 :35 :30 | 3655 | 0.4 | 10 | 32 | 2.3 | 1.38 | 0 | 100 | 0.66 | 0.075 | 5.7 |



**Figure captions**:

Figure 1. (a) Degassing Etna plumes from North East (NEC) and Bocca Nuova (BN) craters as seen from the DLR-Falcon Research aircraft (Photo: Bernadett Weinzierl); (b) Falcon flight track over Sicily on 30 September 2011.

Figure 2. Aircraft observations of Stromboli and Etna plumes on 30 September 2011. Time series of altitude, air temperature ($T$), relative humidity with respect to water ($RH_w$), extinction coefficient ($Ext$) and asymmetry parameter ($g$) are shown by black points. All these parameters are referred to the left-hand y-axes. Time series of HCl, $SO_2$, and $HNO_3$ mixing ratios are shown by blue points and referred to the right-hand y-axes. Shaded areas labelled SV, BN and NEC stand for plume samplings related to Stromboli, Bocca Nuova and North East Etna craters, respectively.

Figure 3. (a) Results of the principal component analysis. First three eigenvectors of the angular scattering intensities (ASI) of the correlation matrix versus measured scattering angles. Values of the first three normalized eigenvalues of the eigenvectors and the remaining variability also displayed. (b) Expansion coefficient diagram: third coefficient versus second coefficient. Black points for cirrus clouds, blue points for contrails, olive points for the North-East-crater plume, violet points for the Stromboly, orange points for BNa and red points for BNb crater plumes. (c) Same as (b) but with the focus of attention on the degassing plumes. Green points for NECa and olive points for NECb crater plumes.

Figure 4. Measured (solid lines) and retrieved (solid black circles) aerosol size distributions (left panels). Measured (solid red circles) and reconstructed (solid black circles) angular scattering intensities (right panels). Stromboly (top), Bocca Nuova (middle) and North East (bottom) Etna craters plums, respectively. (See notations in Fig. 2.)

Figure 5. Probability distribution functions of the asymmetry parameter. Bocca Nuova (BNa, red) and North East (NECb, green) Etna craters plums, respectively. (See notations in Fig. 2.)

Figure 6. Refractive-index dependence of phase functions. The range of the Polar-Nephelometer operating angles is indicated by the gray bar.



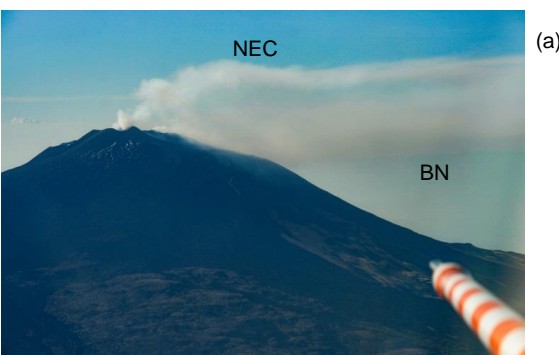

(a)

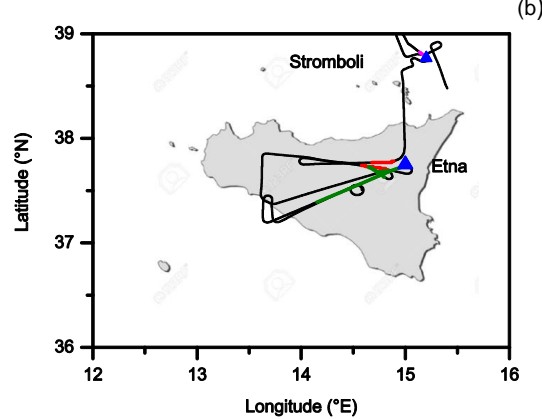

(b)

Fig. 1





Fig. 2





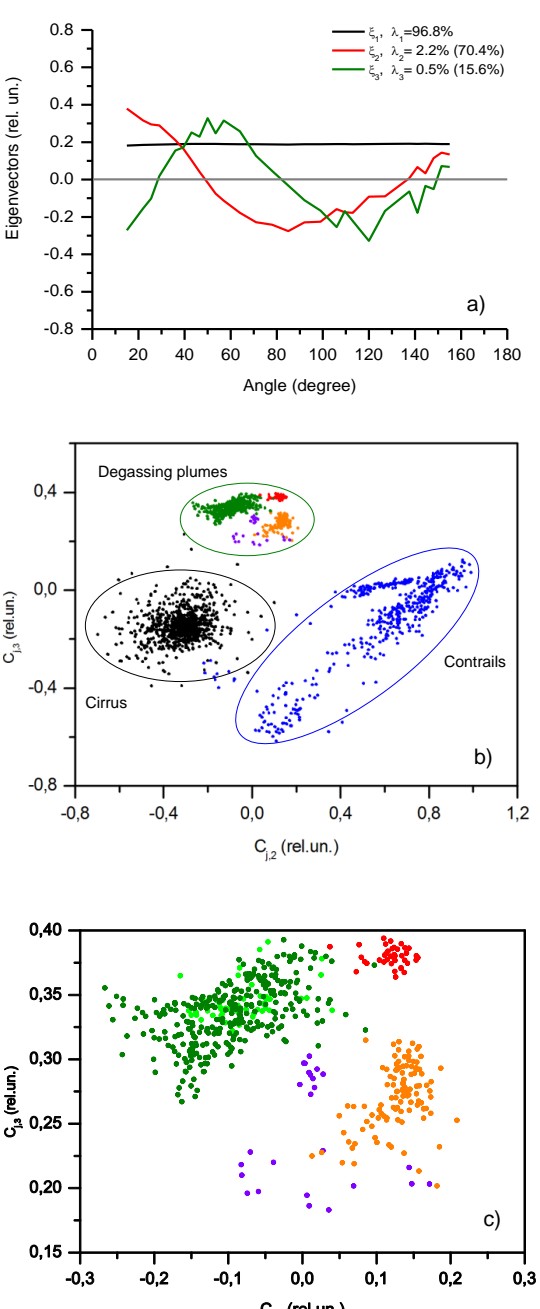

5   Fig. 3





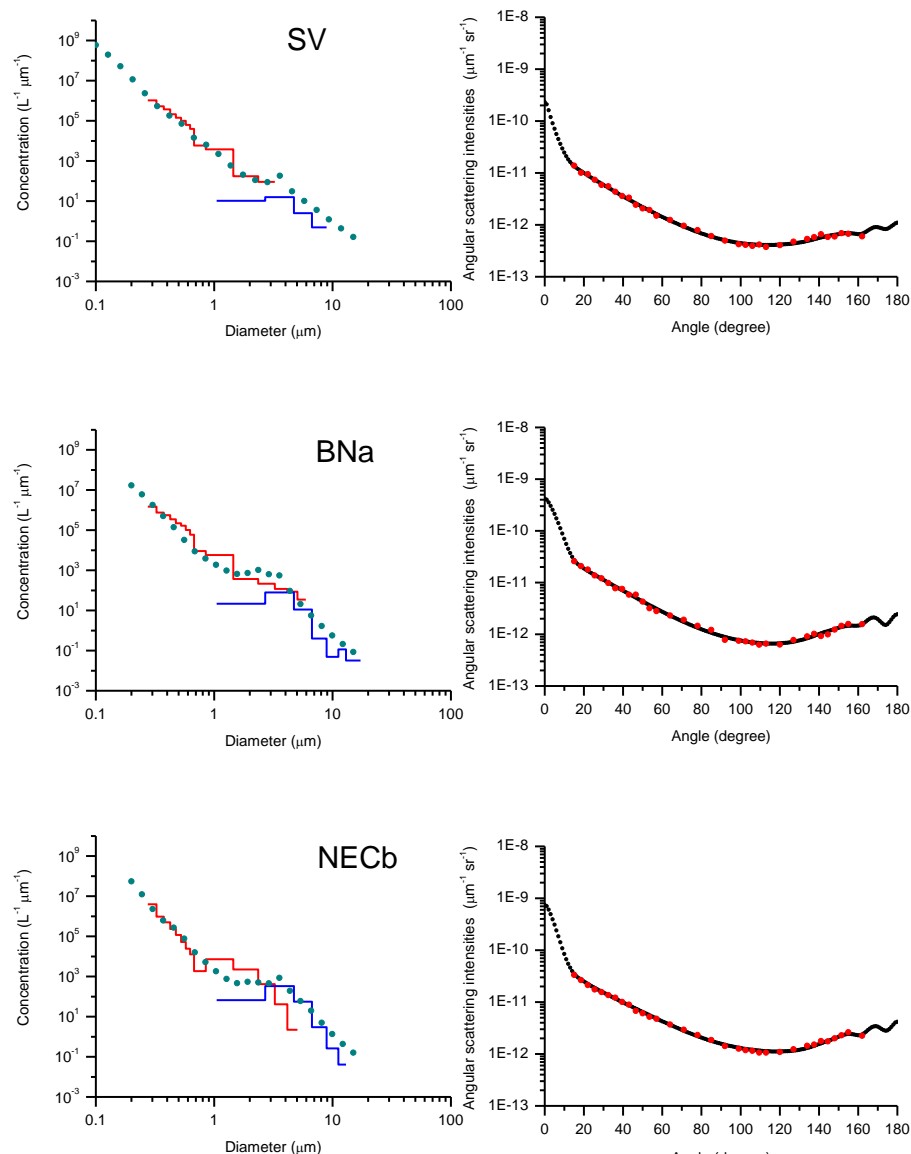

Fig. 4



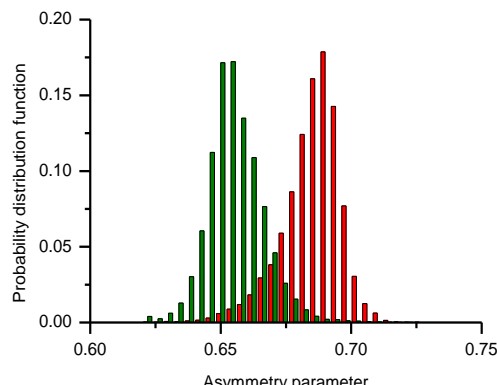

Fig. 5

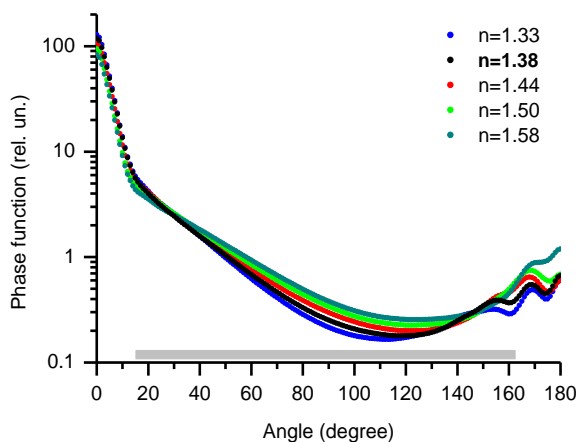

Fig. 6

