# Peer review of "Porous aerosol in degassing plumes of Mt. Etna and Mt. Stromboli"

_Atmospheric Chemistry and Physics, 2016_

## Referee Comment (RC1) · Anonymous Referee #1 · 21 Apr 2016

In my review I focus on the area I am mostly familiar with: retrieval of aerosol properties from the photometric data through computer modeling of light scattering by non-spherical particles. Although the manuscript is well organized and clearly written, there are some concerns which I describe below.

The discussion of the modeling parameters is not sufficient. What was the minimum and maximum size of particles considered? What kind of spheroids (prolate? oblate? aspect ratio?) was used to describe "aspherical particles"? It is also not clear the role of the roughness in the modeling. Importance of roughness is discussed in Section 3 and also presented as one of the parameters of the modeling. However, smooth or rough particles were considered at the modeling is not clear. If smooth, then need to be explained why this assumption was selected. If rough, then characteristics of the roughness should be discussed. Also, what does "severe roughness" mean quantitatively?
As I know, Dubovik's kernels were calculated not for severe roughness but for small roughness parameter equal to 0.2 in the case of spheres and spheroids; for spheres, also the kernels for medium roughness equal to 0.5 were calculated.

The term "aspherical" means a slight deviation from sphericity. As I mentioned above, the characteristics of the spheroids, which represent "aspherical particles" were not mentioned in the paper. However, if spheroids of aspect ratio large than 1.2 were considered, the particles cannot be called "aspherical" and need to be called "non-spherical", although I would highly prefer to replace the word "aspherical" by the word "spheroids" to avoid misunderstandings. If really "aspherical", i.e. particles with aspect ratio close to unity were considered, such a constraint should be justified.

Absolutely not sufficient description is provided for the retrieval procedure and results for size distribution. I could not find any discussion of how this was done, i.e. how the type and parameters of the size distribution were selected and justified. I am also surprised not to see any characteristics of the size distribution in Table 1. As I remember, Dubovik's package works with log-normal size distribution, whereas the plots presented in Fig. 4., left panel, look more like power-law size distribution, moreover, the plots in Fig.4 look like a combination of several power-law distributions, different for different ranges of particles. A more detailed discussion on selecting the size distribution and its final characteristics (size ranges, types and quantitative parameters of the size distribution) is necessary.

I find the discussion about the refractive index insufficient as it considers only its real part. I would expect to see a discussion of the imaginary part of the refractive index too. Is the imaginary part also consistent with the particles of porosity 18-35% ? For this, I would expect the authors to provide the complex refractive index of the material used for the Maxwell Garnett calculations and its justification. Is it consistent with the composition of volcanic aerosols?

I was also confused by the discussion regarding asymmetry (g) parameter. For example, in page 4 the authors claim "Generally, the g-value decreases with increasing asphericity of the particles (Gayet et al., 2002; Gayet et al., 2012)." However, values of asymmetry parameter depend mostly on the size of particles (see, e.g., Asano and Sato, Appl. Optics, 19, 962-974, 1980; or check the plots in http://www.meteo.physik.uni-muenchen.de/~seppg/spheroids.html ) , thus, a discussion of g values and associated with them "asphericity" without identifying the range of particle size is not very useful.

At the end of Section 3.1 a vectorial form of the formula for the principal component analysis is presented. I see it as unnecessary complication which is more confusing than useful. Leaving the

parameters in the form $\xi_l(\theta_i)$, $\ln[\sigma_j(\theta_i)]$ will make them more evident and do not make any conflict with Fig.3, where parameters $\xi_1$, $\xi_2$, $\xi_3$ appeared to be undefined if the vectorial representation is used. Also, please, define parameter $\lambda$ used in Fig. 3.

Minor comments.

Page 3, top line. I highly recommend to add a reference to Mishchenko et al. (Optics Letters, 39, 3935, 2014). Moreover, I think, this paper deserves discussion in the manuscript (e.g., in page 14) as it provides good estimates what is important in modeling heterogeneous particles using effective medium approach. Also, please, fix the typo "Gustafsonm"

Page 9, line 15. What does it mean that the size distribution is not negative?

Page 15, top line, "inertia effect" – what is this ?

---

## Referee Comment (RC2) · Anonymous Referee #2 · 5 May 2016

Manuscript: "Porous aerosol in degassing plumes of Mt. Etna and Mt. Stromboli". By Valery Schcherbakov et al.

Reviewer comments

General comments. Manuscript discusses characterization of volcanic aerosols from Mt. Etna and Mt. Stromboli degassing plumes using both in situ and remote sensing techniques. In situ observations consisted of aerosol size distribution (ASD) measurements using Forward Scattering Spectrometer probes (FSSP) and remote sensing approach was based on inversion of combined observations of angular scattering intensities and extinction obtained by airborne Polar Nephelometer. Information content of Nephelometer observations was analyzed using Principal Component technique which showed possibility to distinguish scattering pattern of volcanic aerosols from the one of

clouds (cirrus and contrails). Inversion of Polar Nephelometer data resulted in relatively low values of the real part of refractive index of volcanic aerosol: 1. 35 to 1.38. This was attributed to the presence of cavities inside particles which effectively decrease the real part of refractive index. Manuscript is very well written and the goals and the techniques used are clear. I believe that the subject of the manuscript is in scope of ACP. Paper certainly can be published. Specific Comments. 1. My main concern is the effect of uncertainty in extinction coefficient (25%) and the limited range in scattering angles (15 to 162) on the accuracy of aerosol retrievals. The authors do not discuss these issues at all. However absence of aureole measurements can affect the ASD retrievals, especially Deff. In addition the uncertainty in extinction coefficient can affect the accuracy of retrieved complex refractive index. Therefore I suggest authors to conduct a simple sensitivity studies: calculate synthetic measurements for the complete range of scattering angles and then invert them using 15-162 range only. In addition, add/subtract 25% to/from extinction coefficient and estimate corresponding uncertainty in retrieved aerosol parameters. I believe these sensitivity tests will make the conclusions of the manuscript much more solid. 2. Did authors really try different initial guesses for inversion code to make sure the global minimum is reached as they discussed at page 9? 3. Is Maxwell Garnett mixing rule really applicable to this type of aerosol particles? How the applicability was estimated and what is the accuracy of estimated air voids? 4. In Table 1., the residuals seem too high for "optically" spherical. It would be interesting to look at the dependence of angular measurements fit as a function of scattering angle.

---

## Author Comment (AC1) · 1 Jul 2016

Response to Referee # 1
We thank the referee for his detailed review and valuable comments. The manuscript has been modified according to the suggestions proposed by the reviewer. The remainder is devoted to the specific response item-by-item of the reviewer's comments:

In my review I focus on the area I am mostly familiar with: retrieval of aerosol properties from the photometric data through computer modeling of light scattering by non-spherical particles. Although the manuscript is well organized and clearly written, there are some concerns which I describe below.

The discussion of the modeling parameters is not sufficient. What was the minimum and maximum size of particles considered? What kind of spheroids (prolate? oblate? aspect ratio?) was used to describe "aspherical particles"? It is also not clear the role of the roughness in the modeling. Importance of roughness is discussed in Section 3 and also presented as one of the parameters of the modeling. However, smooth or rough particles were considered at the modeling is not clear. If smooth, then need to be explained why this assumption was selected. If rough, then characteristics of the roughness should be discussed. Also, what does "severe roughness" mean quantitatively? As I know, Dubovik's kernels were calculated not for severe roughness but for small roughness parameter equal to 0.2 in the case of spheres and spheroids; for spheres, also the kernels for medium roughness equal to 0.5 were calculated.

The application of the Dubovik's package in our work will be described in more details in the supplementary material (see below). To summarize briefly, the spheroid axis ratio $\varepsilon$ is within the range $0.3 \leq \varepsilon \leq 3.0$; the minimum and the maximum sizes of particles belong to the set of assessed parameters; the magnitude $\sigma$ of the surface roughness can take only two values $\sigma = 0.0$ or $\sigma = 0.2$.
The surface roughness is specified by the magnitude $\sigma$ (Yang et al., 2013). According to Yang and Liou (1998), $\sigma = 0.-0.005$, $\sigma = 0.005 - 0.05$, and $\sigma = 0.05 - 0.2$ correspond to slight, moderate, and severe roughness in terms of smoothing corresponding phase functions. In contrast to other definitions of the surface roughness, the magnitude $\sigma$ is related to tilted facets of a particle.

The following text has been added to Section 4.1.
*The aspect-ratio values belong to the range (0.3 – 3.0); the surface-roughness parameter can take only two values 0.0 or 0.2.*

*Details of the software package, we used in this works, as well as of the code application are described in the supplementary material. To summarize briefly, different initial guesses for the inversion code were performed on a multidimensional grid of the input parameters using a number of input files. The minimum and the maximum sizes of particles as well as the spherical/non-spherical partitioning ratio belong to the set of assessed parameters in addition to the refractive index and the size distribution.*

The following text has been added to Section 4.2.
*The following results are especially noteworthy. The real part n of the refractive index belongs to the interval from 1.35 to 1.38; the aerosol particles were either non-absorbing or weakly absorbing; the SNR values are equal to 100 %, i.e., the best fits of the Polar Nephelometer data were obtained with the model of spherical particles (Table 1).*

*The assessed value of the maximal particles diameter is about 15 μm for the all degassing-plume penetrations (Fig. 4). That is, the size parameter of the probed aerosols is rather small (lower than 60 for the wavelength of 0.8 μm). This feature can explain the fact that all our retrieval results were very close for the both value of the surface-roughness parameter. The PN measurements are not sensitive enough to distinguish whether small particles have smooth or rough surface. In view of this result and the SNR values, the degassing-plume aerosols are assumed to be smooth spheres.*

The term "aspherical" means a slight deviation from sphericity. As I mentioned above, the characteristics of the spheroids, which represent "aspherical particles" were not mentioned in the paper. However, if spheroids of aspect ratio large than 1.2 were considered, the particles cannot be called "aspherical" and need to be called "non-spherical", although I would highly prefer to replace the word "aspherical" by the word "spheroids" to avoid misunderstandings. If really "aspherical", i.e. particles with aspect ratio close to unity were considered, such a constraint should be justified.
We followed the review's suggestion. The term "aspherical" was deleted and the term "non-spherical" is employed through the text.

Absolutely not sufficient description is provided for the retrieval procedure and results for size distribution. I could not find any discussion of how this was done, i.e. how the type and parameters of the size distribution were selected and justified. I am also surprised not to see any characteristics of the size distribution in Table 1. As I remember, Dubovik's package works with log-normal size distribution, whereas the plots presented in Fig. 4., left panel, look more like power-law size distribution, moreover, the plots in Fig.4 look like a combination of several power-law distributions, different for different ranges of particles. A more detailed discussion on selecting the size distribution and its final characteristics (size ranges, types and quantitative parameters of the size distribution) is necessary.
In this work, the retrievals were performed with the software package, which was provided us by its authors. The package is well described in the scientific literature (see, e.g., Dubovik et al., 2011; Dubovik et al., 2006), and we employed it as a tool without any scientific contribution. That is why we provide some basic information about the code and details of the code application in our work as the supplementary material.

The following text has been added to Section 4.1.
*Details of the software package, we used in this works, as well as of the code application are described in the supplementary material.*

In particular, we employed the version of the Dubovik's package of the same type that is used in the operational processing of the AErosol RObotic NETwork (AERONET) (see, e.g., Eck et al., 2008). In that version, an aerosol size distribution is presented using size bins; and the size bins are formed with discrete logarithmically equidistant size values (see, e.g., Dubovik et al., 2011, Sections 3 and 5; Dubovik et al., 2006, Sections 2 and 6). No analytical functions like log-normal, gamma, power-law and so on are employed to describe an aerosol size distribution as a whole. Thus, retrieval results are not related to some quantitative parameters of analytical functions. That is why such kind of information is not present in Table 1.

The following text has been added to Section 4.1.

*The aerosol size distribution is presented using size bins; and the size bins are formed with discrete logarithmically equidistant size values.*

I find the discussion about the refractive index insufficient as it considers only its real part. I would expect to see a discussion of the imaginary part of the refractive index too. Is the imaginary part also consistent with the particles of porosity 18-35% ? For this, I would expect the authors to provide the complex refractive index of the material used for the Maxwell Garnett calculations and its justification. Is it consistent with the composition of volcanic aerosols?

We recall that the aerosol particles are found to be either non-absorbing or weakly absorbing according to our retrievals (see Table 1 and Section 5). In addition, the imaginary part $\chi$ of the bulk refractive index of sulfates, nitrates and other inorganic matter presumed to form the degassing plumes is very small, i.e. lower than $10^{-6}$ at the wavelength of 0.8 µm (see, e.g., Gosse et al., 1997). Thus, we assumed that $\chi=0.0$ in our Maxwell Garnett calculations.

The following text has been added to Section 5.

*As for the imaginary part $\chi$, it is very small, i.e. lower than $10^{-6}$ at the wavelength of 0.8 µm (see, e.g., Gosse et al., 1997). We recall that the aerosol particles are found to be either non-absorbing or weakly absorbing according to our retrievals. Thus, we assumed that $\chi=0.0$ in the following estimations.*

I was also confused by the discussion regarding asymmetry (g) parameter. For example, in page 4 the authors claim "Generally, the g-value decreases with increasing asphericity of the particles (Gayet et al., 2002; Gayet et al., 2012)." However, values of asymmetry parameter depend mostly on the size of particles (see, e.g., Asano and Sato, Appl. Optics, 19, 962-974, 1980; or check the plots in http://www.meteo.physik.uni-muenchen.de/~seppg/spheroids.html ) , thus, a discussion of g values and associated with them "asphericity" without identifying the range of particle size is not very useful.

We agree that the asymmetry parameter depends first of all on the size of particles.
The corresponding sentence of Section 2.1 was revised by the following way.

*Generally, the g-value decreases with increasing non-sphericity of the particles all other parameter being the same (Gayet et al., 2002; Gayet et al., 2012).*

At the end of Section 3.1 a vectorial form of the formula for the principal component analysis is presented. I see it as unnecessary complication which is more confusing than useful. Leaving the parameters in the form $\xi_i(\theta_i)$, $\ln[\sigma_j(\theta_i)]$ will make them more evident and do not make any conflict with Fig.3, where parameters $\xi_1$, $\xi_2$, $\xi_3$, appeared to be undefined if the vectorial representation is used. Also, please, define parameter $\lambda$ used in Fig. 3.

The vector form of the formula for the principal components is used in a number of publications of the Laboratoire de Météorologie Physique since 2003 (see, e.g., Jourdan et al., 2003; Jourdan et al., 2010). We prefer keep equations unchanged in order to preserve the coherence with earlier works.
At the same time, we revised the caption to Fig. 3 and defined the parameter λ in the text of Section 3.
The corresponding sentences of the caption to Fig. 3 were revised by the following way.

*First three eigenvectors ($\xi_l$ stands for $\vec{\xi_l}$ ) of the angular scattering intensities (ASI) of the correlation matrix versus measured scattering angles. Values of the first three normalized eigenvalues $\lambda_l$ of the eigenvectors and the remaining variability also displayed.*

The corresponding sentence of Section 3.2 was revised by the following way.
*Figure 3(a) shows the first three principal components along with the corresponding eigenvalues $\lambda_l$ normalized as a percentage of the total variance.*

Minor comments.

Page 3, top line. I highly recommend to add a reference to Mishchenko et al. (Optics Letters, 39, 3935, 2014). Moreover, I think, this paper deserves discussion in the manuscript (e.g., in page 14) as it provides good estimates what is important in modeling heterogeneous particles using effective medium approach.
We prefer to add reference on the comprehensive report by Mishchenko et al., 2016. The following text has been added to Section 1.
*Fundamental aspects and recent developments of the scattering of electromagnetic radiation by a discrete random medium as well as applicability of the EMAs are reported in the work by Mishchenko et al., 2016.*

Also, please, fix the typo "Gustafsonm"
Fixed.

Page 9, line 15. What does it mean that the size distribution is not negative?
As we have underscored in the manuscript, size-distribution retrievals are constrained to be non-negative in the Dubovik code. The question of the nonnegativity constraint is well discussed in the literature related to inverse problems (see, e.g., Liu et al., 1999; Dubovik and King, 2000, Section 4.2.1.1; Fiebig et al., 2005; and references therein).
In brief, a formal solution to an inverse problem can provide negative values for fundamentally positive parameters if there is no nonnegativity constraint. And, such negative values can lead to severe ambiguity, for example, in retrieved aerosol size distributions (see, e.g., Liu et al., 1999).
At the same time, the question of the nonnegativity constraint is beyond the scope of our work. That is why there are no revisions in the text related to it.

Page 15, top line, "inertia effect" – what is this ?
The corresponding sentence of Section 5 was revised according terms by Gallily et al., 1986.
*The volume fraction f has to be considered in aerosols sizing instrumentation based on the inertial separation.*

References (additional to the acp-2016-183 discussion manuscript).

Dubovik, O. and King, M. D.: A flexible inversion algorithm for retrieval of aerosol optical properties from Sun and sky radiance measurements, J. Geophys. Res., 105, 20673–20696, doi:10.1029/2000JD900282, 2000.

Fiebig, M., C. Stein, F. Schröder, P. Feldpausch, and A. Petzold: Inversion of Data Containing Information in the Aerosol Particle Size Distribution Using Multiple Instruments, Journal of Aerosol Sci., 36, 1353-1372, doi:10.1016/j.jaerosci.2005.01.004, 2005.

Gallily, I., D. Schiby, A. H. Cohen, W. Holländer, D. Schless, and D. Stöber: On the Inertial Separation of Nonspherical Aerosol Particles from Laminar Flows. I. The Cylindrical Case, Aerosol Science and Technology, 5:2, 267-286, DOI:10.1080/02786828608959093, 1986.

Gosse, S. F., M. Wang, D. Labrie, and P. Chylek: Imaginary part of the refractive index of sulfates and nitrates in the 0.7–2.6 µm spectral region, Appl. Opt., 36(16), 3622–3634, doi:10.1364/AO.36.003622, 1997.

Liu, Y., W. P. Arnott, and J. Hallett: Particle size distribution retrieval from multispectral optical depth: Influences of particle nonsphericity and refractive index, J. Geophys. Res., 104(D24), 31753–31762, doi:10.1029/1998JD200122, 1999.

Mishchenko, M.I., Dlugach, J.M., Yurkin, M.A., Bi, L., Cairns, B., Liu, L., Panetta, R.L., Travis, L.D., Yang, P., and Zakharova, N.T.: First-principles modeling of electromagnetic scattering by discrete and discretely heterogeneous random media, Phys. Rep., 632, 1–75, doi:10.1016/j.physrep.2016.04.002, 2016.

Yang, P., Liou ,K.N., Single-scattering properties of complex ice crystals in terrestrial atmosphere. Contrib. Atmos. Phys. 71 (2), 223–248, 1998.

*Supplement of*

**Porous aerosol in degassing plumes of Mt. Etna and Mt. Stromboli**

V. Shcherbakov et al.

*Correspondence to*: Valery Shcherbakov (v.shcherbakov@opgc.univ-bpclermont.fr)

The copyright of individual parts of the supplement might differ from the CC-BY 3.0 licence.

**Application of the Dubovik's code**

Detailed information on the Dubovik's code can be found in the works by Dubovik et al. (2006), Dubovik et al. (2011) and references therein. This supplement reproduces some basic information from those publications and underscores details of the code application in our work.

**S1 Kernels for direct and inverse problems**

Direct simulations and retrievals use the concept developed by Dubovik et al. (2006) and model the particles for each size bin as a mixture of spherical and non-spherical aerosol components. The nonspherical component was modeled by ensembles of randomly oriented spheroids (ellipsoids of revolution). Aerosol single-scattering properties are approximated utilizing look-up tables of scattering kernels.

The kernels were computed at grid points. The grid points are equidistant for the real part $n$ of the refractive index, logarithmically equidistant for the spheroid axis ratio, i.e., aspect ratio $\varepsilon$, logarithmically equidistant for the imaginary part $\chi$ of the refractive index, and logarithmically equidistant for particle radius $r$ bins. The size parameter $x = 2\pi r/\lambda$, where $\lambda$ is the wavelength. We employed the code version where those parameters are varying in the following ranges.

$1.29 \leq n \leq 1.696$,

$0.0005 \leq \chi \leq 0.5$,

$0.3 \leq \varepsilon \leq 3.0$,

$0.012 \leq x \leq 626$.

The surface of particles can be smooth $\sigma = 0.0$ or severely rough $\sigma = 0.2$, where $\sigma$ specifies the magnitude of roughness (Yang et al., 2013). According to Yang and Liou (1998), $\sigma = 0. -0.005$, $\sigma = 0.005 - 0.05$ and, $\sigma = 0.05 - 0.2$ correspond to slight, moderate, and severe roughness in terms of smoothing corresponding phase functions.

An advanced version of the look-up tables of scattering kernels was used in the work by Kolokolova et al. (2015) were the rough spheroid model was employed to simulate polarization properties of cosmic dust.

**S2 Input parameters for inverse problem**

The Dubovik's package, we used to retrieve aerosol characteristics, has a large set of input parameters. The following lists present the main subset of the parameters, i.e., the characteristics that are necessary to explain the approach of our supervised retrievals.

The parameters related to the instrumentation and experimental data are the following:
- the number and the values of scattering angles at which the Polar Nephelometer (PN) measures angular scattering intensities (ASIs);
- the measured values of the angular scattering intensities;
- the wavelength of the PN laser beam.

The following parameters are assigned by an operator in the input file of the inversion code.

- the regularization parameter, i.e., the smoothness parameter;

- the number of the bins of the size distributions;
- the minimum particle size;
- the maximum particle size;

- the initial guess for the real part of the refractive index;
- the initial guess for the imaginary part of the refractive index;

- the surface roughness $\sigma$ parameter.

**S3 Output characteristics**

The package, we used in this work, provides the following output characteristics as the retrieval results.

- the residuals;
- the retrieved value of the spherical/non-spherical partitioning ratio (SNR);
- the retrieved value of the real part of the refractive index;
- the retrieved value of the imaginary part of the refractive index;
- the retrieved values of the size distribution;
- the phase function computed for the retrieved size distribution and the refractive index;
- the value of the extinction coefficient computed for the retrieved size distribution and the refractive index.

**S4 Supervised retrievals**

We employed the code version where the surface roughness $\sigma$ is not retrieved and has to be assigned. It can take only two values, i.e., either $\sigma = 0.0$ or $\sigma = 0.2$. Thus, supervised retrievals have to be performed at each value of the surface roughness $\sigma$ individually.

The peculiarity of the inversion code, we used in this work, consists in the fact the executable file works with only one input file and it provides one corresponding output file. In other words, an operator has to prepare a set of input files and has to analyze the corresponding set of output files when the parameters mentioned in Section S2 are varied.

In the following, we try to explain our retrieval approach using programming terms or more specifically, **for**-loops of the C++ Language.

```
for (minimum particle size)
  {
   for (maximum particle size)
     {
       for (regularization parameter)
          {
            for (initial guess for the refractive index)
              {
                  Retrievals
              }
          }
     }
  }
```

where "**for** (parameter)" means that the parameter takes a set of values. The set of values is chosen/assigned by an operator. For example, the set of the regularization parameter is assigned within the range that contains the value corresponding to the "L-curve" method.

It can be seen, inverse problem is solved on the multidimensional grid of the input parameters. The accepted solution to the inverse problem corresponds to the minimum of the residuals.

And, we notice that the minimum and the maximum sizes of particles belong to the set of assessed parameters in addition to the characteristics listed in Section S3.

Generally, the approach is time consuming. On the other hand, it assures that the obtained solution corresponds to the global minimum of an objective function in a generally non-linear case.

**S5 Retrieval-quality assurance**

The final step consists in the verification whether retrieval results are scarcely affected by small variations of the input parameters. The supervised retrievals are performed another time but on the smaller grid of the input parameter and the regularization parameter is varied within a tighter range.

```
for (regularization parameter)
  {
    for (initial guess for the refractive index)
      {
          Retrievals
      }
  }
```

The final solution is considered to be stable, i.e., of good quality, when the retrieved values of the refractive index and of the spherical/non-spherical partitioning ratio are the same; and the variations of the residuals and of the retrieved size distribution are small.

**References**

Dubovik, O., A. Sinyuk, T. Lapyonok, B. N. Holben, M. Mishchenko, P. Yang, T. F. Eck, H. Volten, O. Muñoz, B. Veihelmann, W. J. van der Zande, J-F Leon, M. Sorokin, and I. Slutsker: Application of spheroid models to account for aerosol particle nonsphericity in remote sensing of desert dust, J. Geophys. Res. Atmos., 111, D11208, doi:10.1029/2005JD006619, 2006.

Dubovik, O., Herman, M., Holdak, A., Lapyonok, T., Tanré, D., Deuzé, J. L., Ducos, F., Sinyuk, A., and Lopatin, A.: Statistically optimized inversion algorithm for enhanced retrieval of aerosol properties from spectral multi-angle polarimetric satellite observations, Atmos. Meas. Tech., 4, 975–1018, doi:10.5194/amt-4-975-2011, 2011.

Kolokolova L., H. S. Das, O. Dubovik, T. Lapyonok and P. Yang: Polarization of cosmic dust simulated with the rough spheroid model, Planetary and Space Science 116, 30-38, doi:10.1016/j.pss.2015.03.006, 2015.

Yang, P., Liou ,K.N., Single-scattering properties of complex ice crystals in terrestrial atmosphere. Contrib. Atmos. Phys. 71 (2), 223–248, 1998.

Yang, P., Bi, L., Baum, B.A., Liou, K.-N., Kattawar, G.W., Mishchenko, M.I., and Cole, B.: Spectrally Consistent Scattering, Absorption, and Polarization Properties of Atmospheric Ice Crystals at Wavelengths from 0.2 to 100 μm, J Atmos Sci 70:330–347, doi:10.1175/JAS-D-12-039.1, 2013.

---

## Author Comment (AC2) · 1 Jul 2016

Response to Referee # 2
We thank the referee for his detailed review and valuable comments. The manuscript has been modified according to the suggestions proposed by the reviewer. The remainder is devoted to the specific response item-by-item of the reviewer's comments:

Reviewer comments
General comments.
Manuscript discusses characterization of volcanic aerosols from Mt. Etna and Mt. Stromboli degassing plumes using both in situ and remote sensing techniques. In situ observations consisted of aerosol size distribution (ASD) measurements using Forward Scattering Spectrometer probes (FSSP) and remote sensing approach was based on inversion of combined observations of angular scattering intensities and extinction obtained by airborne Polar Nephelometer. Information content of Nephelometer observations was analyzed using Principal Component technique which showed possibility to distinguish scattering pattern of volcanic aerosols from the one of clouds (cirrus and contrails). Inversion of Polar Nephelometer data resulted in relatively low values of the real part of refractive index of volcanic aerosol: 1. 35 to 1.38. This was attributed to the presence of cavities inside particles which effectively decrease the real part of refractive index. Manuscript is very well written and the goals and the techniques used are clear. I believe that the subject of the manuscript is in scope of ACP. Paper certainly can be published.

Specific Comments.
1. My main concern is the effect of uncertainty in extinction coefficient (25%) and the limited range in scattering angles (15 to 162) on the accuracy of aerosol retrievals. The authors do not discuss these issues at all. However absence of aureole measurements can affect the ASD retrievals, especially Deff. In addition the uncertainty in extinction coefficient can affect the accuracy of retrieved complex refractive index. Therefore I suggest authors to conduct a simple sensitivity studies: calculate synthetic measurements for the complete range of scattering angles and then invert them using 15-162 range only. In addition, add/subtract 25% to/from extinction coefficient and estimate corresponding uncertainty in retrieved aerosol parameters. I believe these sensitivity tests will make the conclusions of the manuscript much more solid.

A large set of sensitivity tests related to the air-borne and laboratory nephelometers of the Laboratoire de Météorologie Physique (LaMP) were performed in the mid-noughties with the participation of the first author of the manuscript. The main attention was paid to cases of angular scattering intensities (ASIs) measured within a limited range of scattering angles. Some results of the tests were reported in the work by Verhaege et al., (2008). It was shown that despite the absence of aureole and backward measurements the real part of the refractive index and the microphysical parameters can be retrieved in the case of the low absorbing particles.
The following text has been added to Section 5.

*That conclusion corroborates with the results of sensitivity tests performed by Verhaege et al., (2008) for ASIs measured within a limited range of scattering angles. It was shown that despite the absence of aureole and backward measurements the real part of the refractive index and the microphysical parameters can be retrieved in the case of the low absorbing particles.*

2. Did authors really try different initial guesses for inversion code to make sure the global minimum is reached as they discussed at page 9?

The application of the Dubovik's package in our work is described in more details in the supplementary material (see also "Response to Referee # 1"). To summarize briefly, different initial guesses for inversion code were performed on a multidimensional grid of the input parameters using a number of input files.
The following text has been added to Section 4.1.

*Details of the software package, we used in this works, as well as of the code application are described in the supplementary material. To summarize briefly, different initial guesses for the inversion code were performed on a multidimensional grid of the input parameters using a number of input files. The minimum and the maximum sizes of particles as well as the spherical/non-spherical partitioning ratio belong to the set of assessed parameters in addition to the refractive index and the size distribution.*

3. Is Maxwell Garnett mixing rule really applicable to this type of aerosol particles? How the applicability was estimated and what is the accuracy of estimated air voids?

The effective-medium approximation (EMA) along with the Maxwell Garnett mixing rule have already been used in a number of works to calculate optical properties of porous particles (see, e.g., Voshchinnikov et al., 2007; Kylling et al., 2014). The question of EMA applicability is discussed in details by Mishchenko et al., 2016.

Our estimation of air voids is given as an interval of values, that is, about 18 to 35 % in terms of the total volume. That interval represents errors (accuracy) of our estimations.
The following text has been added to Section 5.

*The effective-medium approximation along with the Maxwell Garnett mixing rule have already been used in a number of works to calculate optical properties of porous particles (see, e.g., Voshchinnikov et al., 2007; Kylling et al., 2014). The question of EMA applicability is discussed in details by Mishchenko et al., 2016.*

4. In Table 1., the residuals seem too high for "optically" spherical. It would be interesting to look at the dependence of angular measurements fit as a function of scattering angle.

The right panels of Figure 4 show the measured (solid red circles) and the reconstructed (solid black circles) angular scattering intensities (ASIs). The reconstructed (retrieved) ASIs were computed from the retrieved size distribution.
In other words, the dependence of angular measurements fit as a function of scattering angle is shown by the solid black circles at each right panel of Figure 4. It is seen that the measured ASIs are well fitted by the retrieved phase functions.
The following text has been added to the corresponding paragraph.

*In other words, the measured ASIs are well fitted by the retrieved phase functions.*

References (additional to the acp-2016-183 discussion manuscript).

Kylling, A., Kahnert, M., Lindqvist, H., and Nousiainen, T.: Volcanic ash infrared signature: porous non-spherical ash particle shapes compared to homogeneous

spherical ash particles, Atmos. Meas. Tech., 7, 919-929, doi:10.5194/amt-7-919-2014, 2014.

Mishchenko, M.I., Dlugach, J.M., Yurkin, M.A., Bi, L., Cairns, B., Liu, L., Panetta, R.L., Travis, L.D., Yang, P., and Zakharova, N.T.: First-principles modeling of electromagnetic scattering by discrete and discretely heterogeneous random media, Phys. Rep., 632, 1–75, doi:10.1016/j.physrep.2016.04.002, 2016.

Voshchinnikov, N., G. Videen, and T. Henning: Effective medium theories for irregular fluffy structures: aggregation of small particles, Appl. Opt. 46, 4065–4072, doi:10.1364/AO.46.004065, 2007.

---

## Referee Report (RR1)

Review of manuscript ''Porous aerosol in degassing plumes of Mt. Etna and Mt. Stromboli'' by Valery Shcherbakov et al.

This is my second review of the manuscript. Now of the corrected version. Although most of the comments were answered by references on other publications I believe that current version of the manuscript is fully suited for publication in ACP.